# Study protocol for a prospective cohort study to investigate Hemodynamic Adaptation to Pregnancy and Placenta-related Outcome: the HAPPO study

Rianne C Bijl [1], Jérôme M J Cornette,[1] Annemien E van den Bosch,[2] Johannes J Duvekot,[1] Jeroen Molinger,[3,4] Sten P Willemsen,[1,5] Anton H J Koning,[6] Jolien W Roos-Hesselink [2], Arie Franx,[1] Régine P M Steegers-Theunissen,[1] Maria P H Koster [1]

For numbered affiliations see end of article.

**Correspondence to**
Dr Maria P H Koster;
m.p.h.koster@erasmusmc.nl

## ABSTRACT

**Introduction** The importance of cardiovascular health in relation to pregnancy outcome is increasingly acknowledged. Women who develop certain pregnancy complications, in particular preeclampsia, are at higher risk for future cardiovascular disease. Independent of its outcome, pregnancy requires a substantial adaptive response of the maternal cardiovascular system. In the Hemodynamic Adaptation to Pregnancy and Placenta-related Outcome (HAPPO) study, we aim to examine longitudinal maternal haemodynamic adaptation to pregnancy from the preconception period onwards. We hypothesise that women who will develop adverse pregnancy outcomes have impaired cardiovascular health before conception, leading to haemodynamic maladaptation to pregnancy and diminished uteroplacental vascular development.

**Methods and analysis** In this prospective cohort study embedded in the Rotterdam periconception cohort, 200 women with a history of placenta-related pregnancy complications (high-risk group) and 100 women with an uncomplicated obstetric history (low-risk group) will be included. At five moments (preconception, first, second and third trimester and postdelivery), women will undergo an extensive examination of the macrocirculatory and microcirculatory system and uteroplacental vascular development. The main outcome measures are differences in maternal haemodynamic adaptation to pregnancy between women with and without placenta-related pregnancy complications. In a multivariate linear mixed model, the relationship between maternal haemodynamic adaptive parameters, (utero)placental vascularisation indices and clinical outcomes (occurrence of pregnancy complications, embryonic and fetal growth trajectories, miscarriage rate, gestational age at delivery, birth weight) will be studied. Subgroup analysis will be performed to study baseline and trajectory differences between high-risk and low-risk women, independent of subsequent pregnancy outcome.

**Ethics and dissemination** This study protocol was approved by the Medical Ethics Committee of the Erasmus MC, Rotterdam, the Netherlands (MEC 2018–150). Results will be disseminated to the medical community by publications in peer-reviewed journals and presentations at scientific congresses. Also, patient associations will be informed and the public will be informed by dissemination through (social) media.

**Trial registration number** NL7394 (www.trialregister.nl)

### Strengths and limitations of this study

► An integrated approach consisting of parallel non-invasive techniques assessing multiple haemodynamic parameters will compile extensive haemodynamic profiles of participants.
► Haemodynamic adaptation to pregnancy will be measured prospectively from the preconception period onwards to 3 months postdelivery, thereby covering an important window in the reproductive timespan.
► Simultaneous assessment of maternal haemodynamic adaptation to pregnancy and uteroplacental vascular development will provide new insights into the interaction between both mechanisms and the pathophysiology of pregnancy complications.
► Since only multiparous women will be included, conclusions about whether maternal haemodynamic maladaptation is a cause or consequence of previous pregnancy outcome cannot be provided.

## INTRODUCTION
### Context

Placenta-related pregnancy complications (PPC), such as preeclampsia and/or fetal growth restriction (FGR), occur in approximately 10%–15% of all pregnancies.[1–3] PPC affect perinatal morbidity and mortality and significantly impact the long-term cardiovascular health of women and their children with associated healthcare costs for society.[4–6] For example, women with a history of preeclampsia have a twofold to sevenfold increased risk to develop ischaemic heart disease in later life, depending on the

gestational age and severity at onset of preeclampsia.[4] Thus, early identification of women who are at risk for PPC is important to enable subsequent personalised prevention and treatment strategies in order to improve reproductive outcome and to improve lifelong cardiovascular health. To date, most available screening tests for adverse pregnancy outcome use blood pressure and uterine artery blood flow, which reflect the haemodynamic profile of the mother and thereby the likelihood of impaired placental development and subsequent adverse events. Nonetheless, these markers are not directly related to cardiac or placental function, which may explain the moderate predictive performances of most prognostic models.[7 8]

### Haemodynamic adaptation to pregnancy

Pregnancy requires a physiological adaptive response of the maternal cardiovascular system with a substantial rise in plasma volume and cardiac output (CO).[9] Already in the first trimester of pregnancy, a woman's systemic vascular resistance decreases with 35%–40% and her stroke volume increases with 8%.[9] This natural volume overload leads to a reversible, physiological left ventricular dilatation and a possible change in left ventricular diastolic function.[10] In most pregnancies, these changes occur without clinical problems. However, in women who develop PPC, maternal haemodynamic adaptation can be different and diastolic dysfunction is more often present.[11 12]

Moreover, in women with complicated pregnancies, endothelial dysfunction and increased vascular stiffness are also more common.[13 14] The endothelial lining of the vasculature in the human body is of great importance in the regulation of vascular tone. Vascular stiffness increases during normal ageing by the replacement of elastin fibres with less elastic collagen. Endothelial function tests and arterial vascular stiffness measurements have already proven to be of importance in predicting cardiovascular disease.[15]

Together, this all supports the hypothesis that in women who develop PPC an increased risk for cardiovascular disease is unmasked by the stressed state of pregnancy.[16 17] However, knowledge on maternal haemodynamic adaptation to pregnancy, as early as the time of conception, is scarce and often only focuses on one or two parameters instead of the complex interaction between macrocirculatory and microcirculatory parameters. Only a few studies starting prior to pregnancy have shown haemodynamic changes in, for example, blood pressure, heart rate and CO early in pregnancy.[18–22] An integrated approach using parallel techniques and multiple haemodynamic parameters from the preconception period onwards throughout pregnancy is needed to provide insight in individual (patho)physiological haemodynamic adaptation to pregnancy.

### Placental development and function

The adequate establishment of the maternal–fetal vascular interface early in gestation is essential for a

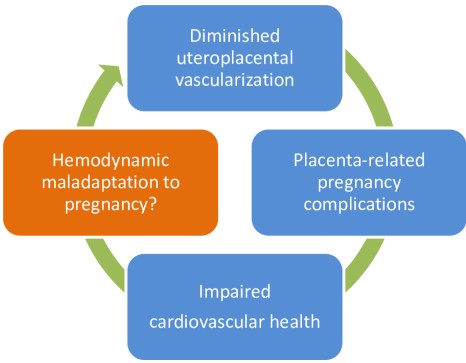

**Figure 1** Schematic hypothesis: Is haemodynamic maladaptation to pregnancy the missing link in the aetiology of placenta-related pregnancy complications?

healthy pregnancy outcome. Among others, this requires adaptation of the uterine spiral arteries into remodelled vessels to create a low-resistance uteroplacental circulation.[23 24] Failure in the adaptation of the uterine vasculature causes hypoperfusion of the placenta and is widely believed to underlie several pregnancy complications.[25 26] This is substantiated by several changes in histopathology, including decidual vasculopathy, villous hypoplasia and lesions consistent with loss of integrity of the vascular wall of the maternal circulation in delivered placentas of women with PPC.[27–29] However, the aetiology of these placental abnormalities and its relation with maternal cardiovascular health remains unclear.

### RESEARCH HYPOTHESIS AND AIM

We hypothesise that impaired maternal cardiovascular health leads to haemodynamic maladaptation to pregnancy, diminished uteroplacental vascular development and subsequent pregnancy complications (figure 1). The overall aim of the Hemodynamic Adaptation to Pregnancy and Placenta-related Outcome (HAPPO) study is to examine associations between maternal haemodynamic adaptation to pregnancy and (utero)placental vascular development in women with and without (a history of) pregnancy complications. Therefore, we specified the following objectives:

1. To assess haemodynamic parameters before, during and after pregnancy to gain insight in (patho) physiological maternal haemodynamic adaptation mechanisms.
2. To assess longitudinal utero(placental) vasculature development using state-of-the-art three-dimensional ultrasound and Virtual Reality techniques combined with in vivo examination of placental bed biopsies.
3. To construct a multivariate model to study the associations between maternal haemodynamic adaptation to pregnancy, utero(placental) vascular development and PPC.
4. To study differences in baseline haemodynamic parameters and haemodynamic adaptation to pregnancy between women with and without a history of PPC.

**Table 1** Inclusion and exclusion criteria for participants in the HAPPO study

| | |
|---|---|
| Inclusion criteria | ► Minimum age of 18 years<br>► Delivery date of last pregnancy more than 1 year ago<br>► Previous pregnancy complicated by PPC *or* uncomplicated previous pregnancy<br>► Current wish to become pregnant |
| Exclusion criteria | ► Unable or unwilling to provide informed consent<br>► Currently breastfeeding |

HAPPO, Hemodynamic Adaptation to Pregnancy and Placenta-related Outcome; PPC, placenta-related pregnancy complications.

## STUDY DESIGN

The HAPPO study is a single-centre prospective cohort study, embedded in the ongoing Rotterdam periconception cohort, performed at the department of Obstetrics and Gynaecology of the Erasmus MC, University Medical Centre Rotterdam, the Netherlands.[30]

## STUDY POPULATION

We aim to include 200 women with a history of pregnancy complications (ie, preeclampsia and/or FGR). Preeclampsia is defined according to the American College of Obstetricians and Gynecologists (ACOG) criteria and FGR is defined conform the Delphi consensus definition described by Gordijn *et al*.[1 31]

Also, 100 women after uncomplicated pregnancies and (term) deliveries will be included as a low-risk control group. Women are eligible to participate in this study when they have a minimum age of 18 years, the delivery date of their last pregnancy was more than 1 year ago, they have a current wish to become pregnant and the previous pregnancy was either complicated by PPC *or* the previous pregnancy was uncomplicated. Women who are unable or unwilling to provide informed consent will be excluded from participation. Also, if a woman is currently breastfeeding, she will be excluded from participation since this

may influence her vascular condition. The inclusion and exclusion criteria are shown in table 1.

## STUDY PROCEDURES/MEASUREMENTS

An overview of all study procedures is provided in figure 2. The first study visit will be planned when the participating woman has the wish to become pregnant, but before conception takes place. When a pregnancy occurs within 1 year, a study visit will be planned in each trimester as well as 3 months postdelivery, which will be the last study visit. All communication with the participants is coordinated by one researcher (RCB) who also performs all study procedures during the study visits.

### Questionnaires

Participants of the study will receive a total of three questionnaires: (1) after the preconceptional study visit, participants will receive an online questionnaire on demographic characteristics, medical conditions and lifestyle; (2) preceding the study visit in the first trimester of pregnancy, participants will receive an online questionnaire to address possible changes in medical conditions and lifestyle; (3) after delivery, participants will receive an online follow-up questionnaire on pregnancy and birth outcomes (ie, gestational age at birth, mode of delivery, birth weight, neonatal sex, pregnancy and/or delivery complications). During the postdelivery visit, the participants will, among others, be asked about current breastfeeding and smoking habits.

### Anthropometrics

At each visit, a woman's height, weight, hip and waist circumference will be measured using standardised, validated methods to compose the classical cardiovascular risk factors body mass index and waist-to-hip ratio.[32]

### Blood sample

Two blood samples (8.5 mL serum tube and 10 mL EDTA tube) will be obtained at the preconception visit, the first-trimester and third-trimester visit and the postdelivery visit. All samples will be centrifuged and divided into aliquots of serum, plasma, whole blood and buffy coat

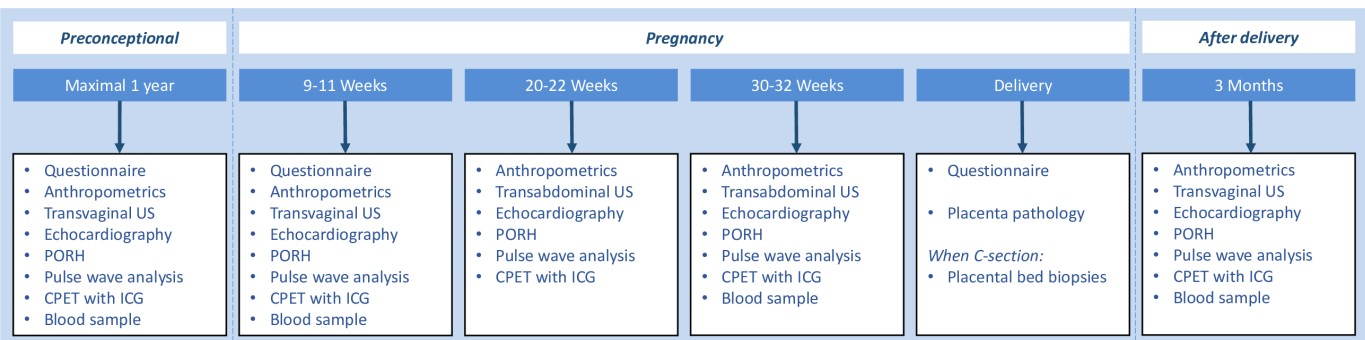

**Figure 2** Overview of study visits and examinations to establish cardiovascular function and utero(placental) vascular development from before conception until after delivery. CPET, cardiopulmonary exercise test; ICG, impedance cardiography; PORH, postocclusion reactive hyperaemia; US, ultrasound.

and then stored in a −80°C freezer at the trial laboratory at our own facility for future assessment of cardiovascular markers (eg, C reactive protein and lipids) and markers of placental function (eg, PlGF and sFlt-1).

### Echocardiography

Transthoracic echocardiography has become the preferential method for safe, non-invasive cardiac assessment in pregnant women.[33 34] At all study visits, echocardiography will be performed using an Affiniti 70G ultrasound machine (Philips Healthcare, Andover, USA) by a trained researcher (RCB) under the supervision of a cardiologist (AEB). Echocardiograms will be fulfilled according to the recommendations of the European Association of Cardiovascular imaging.[35] CO will be derived from the left ventricle outflow tract (LVOT) velocity time integral and LVOT cross-sectional area. From this, systemic vascular resistance will be calculated (ie, mean blood pressure divided by CO). Left ventricular diastolic function will be assessed by mitral inflow (E/A) and tissue Doppler (E/e') examination. Global left ventricular systolic function will be assessed by using the Teichholz method for ejection fraction calculation.[36]

### Postocclusion reactive hyperaemia

One of the first manifestations of an impaired cardiovascular risk profile is endothelial dysfunction of small vessels (microcirculation). Postocclusion reactive hyperaemia will be used in this study to assess the endothelial function of the peripheral microvasculature. We use devices from Moor instruments (moorVMS-VASC 2.0, moorVMS-PRES and moorVMS-LDF; Axminster, UK) to measure microvascular blood flow reactivity following arterial occlusion by using a non-invasive laser Doppler flowmetry (LDF) technique. In resting position, after 3 min of arterial occlusion by a cuff surrounding the upper left arm, the reperfusion of microvascular blood in the skin is measured by the LDF skin probe which is attached to the lower arm. The peak response after the release of the occlusion and the time to recover to baseline reflect endothelial regulated microvascular function.[37 38]

### Pulse wave analysis

A cuff-based oscillometric device (Mobil-O-Graph PWA by I.E.M., Stolberg, Germany) will be used to assess central and peripheral blood pressure, pulse wave velocity and augmentation index (AIx).[39 40] Pulse wave velocity values are derived from an algorithm which integrates age, central systolic blood pressure and data derived from pulse wave analysis into a mathematical model.[41] Higher values of pulse wave velocity indicate higher arterial stiffness and thus a worse vascular condition. The AIx provides additional information on the stiffness of the aorta and is derived from the central pressure waveform.

### Cardiopulmonary exercise test with bioimpedance cardiography

To assess the haemodynamic response during physical stress, CO will also be measured using signal morphology impedance cardiography (SM-ICG) using PhysioFlow PF07 Enduro (Manatec biomedical, Poissy, France). The exercise during which the SM-ICG will be performed is a cardiopulmonary exercise test on a cycle ergometer with a RAMP protocol to 70% of the estimated maximum heart rate according to the Tanaka formula.[42] Moderate exercise limited to 70% of the maximal heart rate is considered safe during pregnancy.[43]

### Uteroplacental vascularisation measurements

Studies using conventional two-dimensional Doppler ultrasound have shown that increased vascular resistance of the uterine arteries is associated with an increased risk for subsequent development of PPC.[7 44] Therefore, pulsed wave Doppler measurements of the right and left uterine artery will be performed at all study visits to provide the pulsatility and resistance indices. Moreover, to evaluate the morphology of the uterine and placental vasculature and quantification of perfusion, a 3D power Doppler scan of the uterine (preconception visit) and placental bed (first-trimester visit) vasculature will be performed, also using the Affiniti 70G ultrasound machine. By assessing these 3D datasets in an innovative virtual reality system (Barco I-Space or 3D desktop), depth perception and three-dimensional interaction enables more accurate and detailed quantification of the utero(placental) vascular volumes. At the Erasmus MC, there is extensive experience with this novel technology and the uteroplacental vascular volume measurements have proven to be feasible and reliable.[45 46]

With respect to the risks of performing ultrasound examinations in the first trimester using 3D power Doppler, recommendations as set by the International Society of Ultrasound in Obstetrics and Gynaecology (ISUOG) are followed for the purpose of this study (ie, thermal index ≤1.0, mechanical index <1.9 and total exposure time <10 min). Transvaginal scanning time (preconception and first trimester) and transabdominal scanning time (second and third trimester) will be kept as short as possible according to the as low as reasonably achievable (ALARA) principles.[47 48]

### Embryonic and fetal growth parameters

At the first trimester visit, additional parameters of embryonic growth will be measured, that is, crown-rump length and embryonic volume. The obtained 3D datasets will be visualised in the Barco I-Space or desktop version and offline volume measurements will be performed.

At the second and third trimester visit, additional parameters of fetal growth (head circumference, abdominal circumference, femur length) and pulsed wave Doppler measurements of the umbilical artery and median cerebral artery will be performed by transabdominal ultrasound.

### Placental examinations

At delivery, histological samples of in vivo placental vasculature will be harvested. The placenta will be collected

according to standard procedures and four samples of approximately $3\,cm \times 3\,cm$ are then fixated in 4% paraformaldehyde for storage. The fixated placenta biopsies will be processed and studied as done in routine placental pathology examination.

In a subgroup of women who undergo a caesarean section (estimated 30% of our study population) at the Erasmus MC, four punch biopsies of the placental bed will be collected, which consist of decidual and myometrial tissue and contain spiral arteries.[49] The samples are directly rinsed with sterile NaCl to wash excessive blood and then fixated in 4% paraformaldehyde for storage and examination. According to our previously published protocol, several markers will be used to score the developmental state of the spiral arteries, identifying typical acute atherosis lesions and evaluating involvement of the immune system.[49 50]

## OUTCOME MEASURES

The primary outcome measure of the HAPPO study is the difference in maternal haemodynamic adaptation to pregnancy, expressed as the trajectory of CO assessed by echocardiography before, during and after pregnancy, between women who do and do not develop PPC. CO is one of the most frequently described haemodynamic parameters in pregnancy, alongside with blood pressure. By using transthoracic echocardiography, as a validated and safe method to measure CO, we will obtain consecutive and reliable information on maternal haemodynamic adaptation to pregnancy.[34]

The secondary outcome measures are differences in additional indices of maternal haemodynamic adaptation to pregnancy and utero(placental) vascular development between women who do and do not develop PPC. This includes differences in weight gain during pregnancy, blood pressure, systemic vascular resistance, left ventricular diastolic function, endothelial regulated microvascular function, pulse wave analysis, haemodynamic response to exercise, placental vascular indices, fetal growth parameters (including Doppler measurements) and placenta/placental bed pathology examinations.

## SAMPLE SIZE CALCULATION

Since we aim to investigate multiple haemodynamic parameters and literature on this subject is scarce, sample sizes are difficult to calculate. There are a few studies that describe the longitudinal trajectory of CO from preconception until after delivery in uncomplicated pregnancies and a couple of cross-sectional studies that investigated CO in either the first of third trimester of pregnancy.[18–20 51 52] Based on these studies, we expect a difference of approximately 0.5 SD at each time point and an intraclass correlation coefficient of at least 0.3 between time points in our study. With alpha set at 0.05 and a power of 0.8, simulation for linear mixed model analysis based on these estimates

showed that 38 cases (PPC in the anticipated pregnancy) are needed to establish such differences.[53]

Based on literature, we expect a 25% recurrence rate of PPC in the high-risk group and a 2% incidence of PPC in the low-risk group.[54–56] Thus, PPC cases will primarily occur in the high-risk group. For an expected number of 38 cases, we would need to follow-up 38/0.25=152 pregnancies in the high-risk group.

Multiparous women who wish to become pregnant who are actively contemplating pregnancy are assumed to have an approximately 85% chance to become pregnant again within 1 year.[57 58] The overall miscarriage rate in this study group is estimated to be approximately 10%.[59] With this taken into account, we would need to include 152/(0.85*0.90)=198 high-risk women prior to conception. For comparison with women at low risk for PPC and to increase the number of controls, we will also include 100 low-risk women with an uncomplicated previous pregnancy. With this sample size, we expect, based on previous research, to be able to detect differences of approximately 5%–10% in other haemodynamic parameters and utero(placental) vasculature development (secondary outcomes).[11 14 18 51 60–66]

## STATISTICAL ANALYSIS PLAN

Patient characteristics, medical conditions and lifestyle data will be presented as means (SD) or numbers (percentage) and compared using t-tests and $\chi^2$ tests. Haemodynamic parameters will be presented as means (SD) at each time point and compared between groups using t-tests.

Multivariate linear mixed models will be used to analyse the trajectories of maternal haemodynamic parameters over time. We will test whether there is a difference between women who do and do not develop PPC by comparing each trajectory using a likelihood ratio test.

Subgroup analyses will be performed to study (baseline and trajectory) differences between high-risk and low-risk women, independent of subsequent pregnancy outcome. Bonferroni corrections will be applied to control for multiplicity of secondary outcome parameters.

In case of significant differences in patient characteristics between groups, regression analyses will be adjusted for these characteristics.

## DATA COLLECTION, HANDLING AND STORAGE

The study will be performed according to written informed consent procedures which ensures personal data protection and confidentiality. Data will be deidentified (pseudoanonymised) in order to guarantee the privacy of the participants, and the key to the code will be safeguarded by the investigators and will remain on site with the local research team.

All research data will be retained and stored in the HAPPO-database at a secured server at the research institution and it will remain there for at least 15 years after

completion of the project. All imaging data will be stored and handled in the same way and according to the same rules and regulations as other patient-related imaging data at Erasmus MC.

Monitoring of data and safety will be performed once a year, according to local protocol for research with negligible risk.

## PATIENT AND PUBLIC INVOLVEMENT

Patient and public were not involved in the design of the study. However, at the earliest moment of recruitment, two large patient associations for women who experienced PE and/or HELLP syndrome (ie, HELLP Stichting (http://www.hellp.nl) and the Preeclampsia Foundation (http://www.preeclampsia.org)) were informed and are involved in promoting the HAPPO study.

## ETHICS AND DISSEMINATION

The study will be conducted according to the principles of the Declaration of Helsinki (version October 2013) and in accordance with relevant national guidelines, regulations and Acts (eg, the Medical Research Involving Human Subjects Act (WMO)). The study protocol was approved by the Medical Ethics Committee of the Erasmus MC (MEC-2018–150). The study is registered in the Dutch Trial Registry (NL7394).

The HAPPO study team invites colleagues who are interested in collaboration or data sharing to contact the principal investigators at HAPPO.studie@erasmusmc.nl.

Results of the study will be disseminated to healthcare professionals and to scientific and industrial peers through events, professional organisations and publications in peer-reviewed (preferably open-access) journals as well as presentations at scientific congresses. Also, results will be communicated to women who wish to become pregnant and women who experienced pregnancy complications by informing patient associations and by dissemination via (social) media.

## FUTURE IMPLICATIONS

The results of this study will provide more understanding of longitudinal maternal haemodynamic adaptation to pregnancy in relation to utero(placental) vascular development and its associations with subsequent pregnancy outcome. If we succeed to establish an association between maternal haemodynamic adaptation to pregnancy and subsequent pregnancy outcome, a larger cohort study in nulliparous women will be needed to investigate causal relationships between haemodynamic adaptation to pregnancy and placenta-related outcomes. Hereby, starting points for future possibilities to optimise cardiovascular and placental health by interventions in clinical validation studies will arise. This will enable the development of more accurate, personalised prevention and treatment strategies for women with high-risk pregnancies, from the preconception period onwards. Thereby, we aim to reduce subsequent maternal and perinatal morbidity and mortality in the future, eventually leading to life-long reduced risks and costs of cardiovascular diseases.

**Author affiliations**
[1]Department of Obstetrics and Gynecology, Erasmus MC, Rotterdam, The Netherlands
[2]Department of Cardiology, Erasmus MC, Rotterdam, The Netherlands
[3]Department of Intensive Care Adults, Erasmus MC, Rotterdam, The Netherlands
[4]Department of Anesthesiology & Intensive Care Medicine, Human Physiology and Pharmacology Lab (HPPL), Duke Medicine, Durham, North Carolina, USA
[5]Department of Biostatistics, Erasmus MC, Rotterdam, The Netherlands
[6]Department of Pathology, Erasmus MC, Rotterdam, The Netherlands

**Contributors** The rationale and design of the study were conceived by JMJC, JJD, JWRH, RST and MPHK. AEB, JM and AHJK contribute their invaluable practical expertise and SPW his excellent statistical expertise. RCB is the executive researcher, supervised by JMJC, AF and MPHK. MPHK is the principle investigator of the study and responsible for all aspects of the study. RCB and MPHK prepared the first draft of the protocol. All authors have read and approved the final manuscript.

**Funding** This work is supported by an Erasmus MC Fellowship Grant (2017) and a Peter Joseph Pappas Research Grant from the Preeclampsia Foundation (2019), both awarded to MPHK.

**Competing interests** None declared.

**Patient consent for publication** Not required.

**Provenance and peer review** Not commissioned; externally peer reviewed.

**ORCID iDs**
Rianne C Bijl http://orcid.org/0000-0002-0914-3206
Jolien W Roos-Hesselink http://orcid.org/0000-0002-6770-3830
Maria P H Koster https://orcid.org/0000-0002-4786-0361

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
