## [Reviewer comments · BMJ Open]

ARTICLE DETAILS

TITLE (PROVISIONAL)	A study protocol for a prospective cohort study to investigate Hemodynamic Adaptation to Pregnancy and Placenta-related Outcome: the HAPPO study
AUTHORS	Bijl, Rianne; Cornette, Jérôme; van den Bosch, Annemien; Duvekot, Johannes; Molinger, Jeroen; Willemsen, Sten; Koning, Anton; Roos-Hesselink, Jolien; Franx, Arie; Steegers-Theunissen, Régine; Koster, Maria PH

VERSION 1 - REVIEW

REVIEWER	Tullio Ghi University of Parma Italy
REVIEW RETURNED	04-Aug-2019

GENERAL COMMENTS	This is a very interesting research project aimed at assessing the role of maternal cardiovascular maladaptation in the origin of placental mediated pregnancy complications The robust sample size of the study, the innovative and accurate tools for patients assessment together with the longitudinal evaluation of the subjects starting from before conception throughout the pregnancy to the postpartum period are certainly major strengths of this project I have hence two major concerns regarding the study design 1. The Authors have decided to include in the high risk group women with history preeclampsia and those with a previous SGA neonate (defined as birthweight below the 10th centile). As they are interest to select a group of women at risk for placental disorders and they claim to investigate the effects of these disorders on maternal cardiac adaptation to the following pregnancy I wonder why the did not decide to include only women whose previous pregnancy was complicated by IUGR (fetal growth restriction) according to the widely accepted criteria. This group is certainly more closely related to a placental disorder compared with cases of SGA fetuses which can be due to constitutionally factors. 2. As the pregnancy itself is known to have an impact on maternal heart, this study will not be able to spot the true association between cardiac findings and pregnancy outcome or placental function. This project will give us a clue on the effect of a previous pregnancy on maternal cardiac function during the subsequent pregnancy and will try to assess if a history of pregnancy complicated by a placental mediated disorder will be more prone to affect cardiac adaptation to the following pregnancy and to increase the risk of pregnancy complications. On te basis of these
---

	considerations the main objective of this research should be modified accordingly
--	---

REVIEWER	C. Severens-Rijvers Maastricht University Medical Center+ The Netherlands
REVIEW RETURNED	02-Sep-2019

GENERAL COMMENTS	The authors have developed an interesting study protocol. I am looking forward to read the results. I have a few questions for the authors: - Have you considered loss to follow-up in your sample size calculations? - Will you ask if a mother is breastfeeding during the postpartum measurement? I would suggest to control for breastfeeding in the postpartum measurements.
--

VERSION 1 – AUTHOR RESPONSE

Reviewer: 1

General:

This is a very interesting research project aimed at assessing the role of maternal cardiovascular maladaptation in the origin of placental mediated pregnancy complications. The robust sample size of the study, the innovative and accurate tools for patients assessment together with the longitudinal evaluation of the subjects starting from before conception throughout the pregnancy to the postpartum period are certainly major strengths of this project. I have hence two major concerns regarding the study design:

1. The Authors have decided to include in the high risk group women with history preeclampsia and those with a previous SGA neonate (defined as birthweight below the 10th centile). As they are interest to select a group of women at risk for placental disorders and they claim to investigate the effects of these disorders on maternal cardiac adaptation to the following pregnancy I wonder why they did not decide to include only women whose previous pregnancy was complicated by IUGR (fetal growth restriction) according to the widely accepted criteria. This group is certainly more closely related to a placental disorder compared with cases of SGA fetuses which can be due to constitutionally factors.

We thank the reviewer for the valuable input that helped us to further improve our manuscript. Our answers (in bold) are as follows:

We agree with the reviewer that constitutional small for gestational age (SGA) neonates are not necessarily a consequence of placental-mediated pregnancy complications and including women with a history of SGA, without intra-uterine growth restriction, may possibly alter our findings. The reason for this inclusion criterion was mainly practical: due to registration issues, the foetal growth trajectory

during a previous pregnancy is not always known to the investigators (while birthweight and birthweight percentile are). However, upon your suggestion we have decided to narrow our inclusion criteria to create a true high risk group in which we can investigate hemodynamic adaptation to pregnancy. In the high risk group, we will include women with a history of foetal growth restriction conform the Delphi consensus definition and added the reference of Gordijn et al. We have adjusted the HAPPO trial registry details (<https://www.trialregister.nl/trial/7394>) and have adjusted the paragraph on the study population in the manuscript to clarify this (page 7, line 16-19):

“We aim to include 200 women with a history of pregnancy complications (i.e. preeclampsia and/or foetal growth restriction [FGR]). Preeclampsia is defined according to the American College of Obstetricians and Gynecologists (ACOG) criteria and FGR is defined conform the Delphi consensus definition described by Gordijn et al. (Reference: Gordijn et al., 2016, Consensus definition of fetal growth restriction: a Delphi procedure)”

2. As the pregnancy itself is known to have an impact on maternal heart, this study will not be able to spot the true association between cardiac findings and pregnancy outcome or placental function. This project will give us a clue on the effect of a previous pregnancy on maternal cardiac function during the subsequent pregnancy and will try to assess if a history of pregnancy complicated by a placental mediated disorder will be more prone to affect cardiac adaptation to the following pregnancy and to increase the risk of pregnancy complications. On the basis of these considerations the main objective of this research should be modified accordingly

In accordance to the reviewer’s comment, the HAPPO study will indeed not prove any causal relationship between maternal cardiovascular maladaptation to pregnancy and placenta-related pregnancy complications. Our main objective states: to study associations between maternal hemodynamic adaptation to pregnancy and (utero)placental vascular development in women with and without pregnancy complications. By emphasizing that this is an observational association study we think that we refrain from claiming to study any causal relations.

Furthermore, as stated in the fourth specified objective, we will not only investigate associations between hemodynamic adaptation to the subsequent pregnancy and new onset placenta-related complications, but we will also study differences in baseline parameters and hemodynamic adaptation to the subsequent pregnancy between high and low risk women, independent of pregnancy outcome. For clarification, we have added “(a history of)” in the overall objective statement (page 6 line 26-31, page 7 line 1-8):

“The overall aim of the HAPPO study is to examine associations between maternal hemodynamic adaptation to pregnancy and (utero)placental vascular development in women with and without (a history of) pregnancy complications. Therefore, we specified the following objectives:

1. To assess hemodynamic parameters before, during and after pregnancy to gain insight in (patho)physiological maternal hemodynamic adaptation mechanisms;
2. To assess longitudinal utero(placental) vasculature development using state-of-the-art three-dimensional ultrasound and Virtual Reality techniques combined with in vivo examination of placental bed biopsies;

3. To construct a multivariate model to study the associations between maternal hemodynamic adaptation to pregnancy, utero(placental) vascular development and PPC.
4. To study differences in baseline hemodynamic parameters and hemodynamic adaptation to pregnancy between women with and without a history of PPC.”

If we succeed to establish associations between maternal hemodynamic adaptation to pregnancy and subsequent pregnancy outcomes, a larger cohort study in nulliparous women will be needed to investigate the hemodynamic adaptation to the first pregnancy of a woman. This would be the next step towards early recognition of women at risk for pregnancy complications and possible prevention of placenta-related pregnancy complications. We have amended the paragraph on future implications accordingly (page 14 line 30-31, page 15 line 1-10):

“The results of this study will provide more understanding of longitudinal maternal hemodynamic adaptation to pregnancy in relation to utero(placental) vascular development and its associations with subsequent pregnancy outcome. If we succeed to establish an association between maternal hemodynamic adaptation to pregnancy and subsequent pregnancy outcome, a larger cohort study in nulliparous women will be needed to investigate causal relationships between hemodynamic adaptation to pregnancy and placenta-related outcomes. Hereby, starting points for future possibilities to optimize cardiovascular and placental health by interventions in clinical validation studies will arise. This will enable the development of more accurate, personalized prevention and treatment strategies for women with high-risk pregnancies, from the preconception period onwards. Thereby, we aim to reduce subsequent maternal and perinatal morbidity and mortality in the future, eventually leading to life-long reduced risks and costs of cardiovascular diseases.”

Reviewer: 2

General:

The authors have developed an interesting study protocol. I am looking forward to read the results.

I have a few questions for the authors:

1. Have you considered loss to follow-up in your sample size calculations?

We thank the reviewer for the valuable input that helped us to further improve our manuscript. Our answers (in bold) are as follows:

In our sample size calculation we did not specifically take a predetermined loss to follow-up rate into account. From previous studies at our department (for example the VIRTUAL Placenta study, which is also embedded in the Rotterdam Periconception Cohort) we know that if a small research team is involved and participants have regular study visits, the loss to follow-up rate is quite low (<5%). In the HAPPO study, communication with the participants is coordinated by one researcher (Bijl); this is also

the researcher who performs all study procedures during all study visits. Therefore, close monitoring of follow-up of participants is guaranteed and well arranged. We have added this to the manuscript for clarification (page 8, line 7-9):

“All communication with the participants is coordinated by one researcher (RCB) who also performs all study procedures during the study visits.”

2. Will you ask if a mother is breastfeeding during the postpartum measurement? I would suggest to control for breastfeeding in the postpartum measurements.

Breastfeeding is indeed a known mediator of cardiovascular measurements and therefore women who are currently breastfeeding are excluded from participation in the HAPPO study. Also, we ask for current breastfeeding at the post-delivery study visit to be able to adjust for this parameter in our statistical analysis. For the same reason, we ask for current smoking habits at the post-delivery study visit. We have now clarified this in the main text of our manuscript (page 8, line 18-20):

“During the post-delivery visit, the participants will, among others, be asked about current breastfeeding and smoking habits.”

VERSION 2 – REVIEW

REVIEWER	Carmen Severens-Rijvers Maastricht University Medical Center+, The Netherlands
REVIEW RETURNED	21-Oct-2019
GENERAL COMMENTS	The reviewer completed the checklist but made no further comments.